# Case-Fatality and Temporal Trends in Patients with Psoriasis and End-Stage Renal Disease

**DOI:** 10.3390/jcm11154328

**Published:** 2022-07-26

**Authors:** Johannes Wild, Karsten Keller, Susanne Karbach, Julia Weinmann-Menke, Thomas Münzel, Lukas Hobohm

**Affiliations:** 1Department of Cardiology, Cardiology I, University Medical Center Mainz, Johannes Gutenberg University Mainz, 55131 Mainz, Germany; johannes.wild@unimedizin-mainz.de (J.W.); karsten.keller@unimedizin-mainz.de (K.K.); karbasu@uni-mainz.de (S.K.); 2Center for Thrombosis and Hemostasis (CTH), University Medical Center Mainz, Johannes Gutenberg University Mainz, 55131 Mainz, Germany; tmuenzel@uni-mainz.de; 3Medical Clinic VII, Department of Sports Medicine, University Hospital Heidelberg, 69120 Heidelberg, Germany; 4German Center for Cardiovascular Research (DZHK), Partner Site Rhine-Main, 55131 Mainz, Germany; 51st Department of Medicine, Johannes Gutenberg-University Mainz, 55131 Mainz, Germany; julia.weinmann-menke@unimedizin-mainz.de

**Keywords:** ESRD, psoriasis, mortality, trends, haemodialysis

## Abstract

Background and Objectives: During the last decades, growing evidence corroborates that chronic inflammatory disease impairs the body beyond the cutaneous barrier. Linkage between psoriasis and kidney disease, and in particular between psoriasis and end-stage renal disease (ESRD), have not yet been elucidated. We sought to analyze the impact of concomitant psoriasis on the in-hospital outcomes of patients hospitalized with ESRD. Patients and Methods: We analyzed data on characteristics, comorbidities, and in-hospital outcomes of all hospitalized patients with ESRD stratified for concomitant psoriasis in the German nationwide in-patient sample between 2010 and 2020. Results: Overall, 360,980 hospitalizations of patients treated for ESRD in German hospitals were identified from 2010 to 2020 and among these 1063 patients (0.3%) additionally suffered from psoriasis. While the annual number of all ESRD patients increased within this time, the number of patients with ESRD and the additional psoriasis diagnosis decreased slightly. Patients with ESRD and psoriasis were five years younger (66 [IQR, 56–75] vs. 71 [59–79] years, *p* < 0.001), were more often obese (17.5% vs. 8.2%, *p* < 0.001) and more frequently had cancer (4.9% vs. 3.3%, *p* < 0.001), diabetes mellitus (42.7% vs. 38.5%, *p* = 0.005) and coronary artery disease (31.1% vs. 28.0%, *p* = 0.026). Multivariate regression models demonstrated that psoriasis was not associated with in-hospital case-fatality in patients with ESRD (OR 1.02 (95%CI 0.78–1.33), *p* = 0.915). Conclusions: ESRD patients with the concomitant psoriasis diagnosis were hospitalized on average 5 years earlier than patients without psoriasis. A higher prevalence of severe life-shortening comorbidities including coronary artery disease and cancer was detected in ESRD patients with psoriasis despite their younger age. Our findings support the understanding of psoriasis as an autoimmune skin disease crossing the boundary between dermatology and internal medicine.

## 1. Introduction

Over a long period of time, the skin disease psoriasis was an exclusive domain of dermatologists. The visible signs of the Interleukin-17A driven autoimmune disease affect the body surface quite obviously. However, in the last decades, the understanding of psoriasis, which is one of the most common skin diseases with prevalence between 2% and 6% in industrialized countries [1,2], has changed fundamentally. Evidence is accumulating that psoriasis is a chronic multisystem disease associated with a variety of concomitant diseases [3]. The term “psoriatic march” reflects this systemic character of the disease and creates a pathophysiological link between skin and systemic inflammation [4]. According to this concept, starting from chronic skin inflammation, systemic inflammation follows, leading to vascular dysfunction and insulin resistance, and in further progression to atherosclerosis and its acute manifestations, such as myocardial infarction and stroke [5]. These concomitant diseases did not only cause a reduced quality of life, but also reduced life expectancy of these patients, who die approximately three to six years earlier compared to patients without psoriasis [6]. Therefore, the development of preventive strategies aiming at the cardiovascular comorbidity has been considered as crucial, and implementation of these strategies has been envisaged for psoriasis patients [7]. Both clinical and basic research has begun to provide evidence for pathophysiological connections between autoimmune skin and heart disease [8,9]. While the strong association between cardiovascular disease and psoriasis seems clearly provable both epidemiologically and mechanistically, the view is different concerning concomitant chronic kidney disease. Whereas several publications provide evidence for an increased incidence of chronic kidney disease (CKD) with psoriasis [10,11,12], the pathophysiological relationship between the two diseases remains controversial [13,14]. Vice versa, patients with chronic kidney disease had higher risk for the development of psoriasis [15]. Whereas there is evidence for a significantly increased risk of death associated with kidney diseases in psoriasis patients, a cause-and-effect relationship has not been established. Moreover, the impact of psoriasis on outcomes in different stages of kidney disease is not clear, but this is crucial because the various stages of chronic kidney disease (CKD) differ widely regarding mortality [16,17]. As patients with end-stage renal disease (ESRD) have by far the highest mortality of all CKD patients, we analyzed comorbidities and in-hospital outcomes and events in hospitalized patients with ESRD and concomitant psoriasis compared to patients without psoriasis.

## 2. Patients and Methods

### 2.1. Source of Data and Diagnoses Codes

The Research Data Center (RDC) of the Federal Statistical Office and the Statistical Offices of the federal states in Wiesbaden, Germany (source: RDC of the Federal Statistical Office and the Statistical Offices of the federal states, DRG Statistics 2010–2020, own calculations), provided us the data as aggregated statistical results on the basis of SPSS codes (SPSS^®^ software, version 20.0, SPSS Inc., Chicago, IL, USA), which were previously sent by us to the RDC [18,19]. In-hospital diagnoses (German Diagnosis Related Groups (G-DRG)) together with diagnostic, surgical, and interventional procedures (OPS, surgery and procedure codes (Operationen- und Prozedurenschlüssel)) are coded in Germany and other countries according to the International Classification of Diseases and Related Health Problems, 10th Revision with German Modification (ICD-10-GM). All DRG diagnoses and OPS codes of the hospitalized patients are gathered by the Federal Statistical Office of Germany.

### 2.2. Study Parameters, Outcomes, and Ethical Statement

For the present analysis, we identified and included all hospitalized patients with ESRD defined as GFR below 15 mL/min/1.73m^2^ by the ICD-code N18.5 between the years 2010 and 2020. The included hospitalizations of patients with ESRD were stratified for the presence of psoriasis (ICD-code L40).

The primary end point of this study was defined as death from any cause during hospitalization (all-cause in-hospital death). The second endpoint focused on an aggravated cardiovascular profile including: (I) serious cardiovascular events during hospitalization, such myocardial infarction (ICD-codes I21 and I22), ischemic stroke (ICD-code I63), and pulmonary embolism (ICD-code I26); (II) cardiovascular risk factors such as diabetes mellitus (ICD-codes E10-E14), essential arterial hypertension (ICD-code I10), and obesity (defined as body mass index (BMI) ≥ 30 kg/m^2^ according to the World Health Organization; ICD-code E66); and (III) cardiovascular comorbidities such as coronary artery disease (ICD-code I25), left- and right-heart failure (I50.0 and I50.1), and atrial fibrillation/flutter (ICD-code I48). In addition, necessity for transfusion of erythrocyte concentrates (OPS code 8–800) and clinically relevant bleeding events, such as intracerebral bleeding (ICD-code I61) or gastrointestinal bleeding (ICD-code IK920, K921, und K922)), were detected and analyzed.

Because this study did not involve direct access to individual patients’ data by the investigators, approval by an ethics committee and informed consent were not required under German law.

### 2.3. Methods for Statistical Analyses

Temporal trends of annual and age-related hospitalizations of ESRD patients with and without psoriasis together with relative mortality rate (case-fatality rate) were calculated on an annual and age-dependent (age-decade) basis. Linear regressions were used to assess trends over time and the results are shown as beta (β) with corresponding 95% confidence intervals (CI). We compared hospitalized patients with ESRD and with or without concomitant psoriasis and survivors vs. nonsurvivors in ESRD patients with psoriasis by using the Fisher’s exact or chi-square test for categorical variables and the Mann–Whitney U test for continuous variables. Univariate and multivariate logistic regression models were performed to examine the influence of comorbidities, age, and clinical conditions on in-hospital mortality. We presented the results as odds ratios (OR) and corresponding 95% CIs. To demonstrate statistical independence of the aforementioned parameters, multivariate logistic regression models were used by including the following parameters for adjustment: sex, age, cancer (ICD-codes C00-C97), coronary artery disease (ICD-code I25), heart failure (ICD-code I50), chronic obstructive pulmonary disease (COPD, ICD-code J44), essential arterial hypertension, and diabetes mellitus.

For statistical analysis, we used the SPSS^®^ software, version 20.0 (SPSS Inc., Chicago, IL, USA), and considered *p* values of <0.05 (two-sided) to be statistically significant.

## 3. Results

### 3.1. Patient Collective

The German nationwide inpatient sample included 360,980 hospitalizations of patients with ESRD and 47,116 with psoriasis between the years 2010 and 2020; 1063 (0.3%) of the ESRD hospitalizations were coded with both ESRD and psoriasis. Most patients were male (59.8%) (Figure 1A) with a median age of 70 (IQR 58–78) years. ESRD patients with concomitant psoriasis were five years younger than those without psoriasis (66 [IQR, 56–75] vs. 71 [59–79] years, *p* < 0.001), were more often obese (17.5% vs. 8.2%, *p* < 0.001), and suffered more frequently from comorbidities such as cancer (4.9% vs. 3.3%, *p* < 0.001), diabetes mellitus (42.7% vs. 38.5%, *p* = 0.005), and coronary artery disease (31.1% vs. 28.0%, *p* = 0.026). Table 1 gives an overview of the demographic and patient characteristics of the two groups.

### 3.2. In-Hospital Trends over Eleven Years

Annual numbers of hospitalizations of ESRD patients increased from 30,149 cases in 2005 to 32,022 cases in 2016, whereas the annual numbers of psoriasis patients stayed almost constant from 2775 cases in 2005 to 2787 cases in 2020. Focusing on the patient numbers with ESRD and psoriasis during our 10-year observational timeframe, the hospitalizations of ESRD patients with psoriasis decreased slightly from 101 cases in 2010 to 78 cases in 2020. The case-fatality rate decreased during the same period from 5.9% in 2010 to 3.8% in 2020 (β −0.53 [95%CI −1.34 to −0.29]; *p* = 0.204) but grew exponentially with age. However, although the case-fatality rate increased with age from the third decade to the eight decade, it decreased in patients over the age of 80 (Figure 1B,C). In contrast, comorbidities as cancer (β 0.61 [95%CI −0.24 to 1.47], *p* = 0.161), left-heart failure (β 0.59 [95%CI 0.18 to 1.01], *p* < 0.005), atrial fibrillation/flutter (β 0.18 [95%CI −0.27 to 0.62], *p* = 0.437), or obesity (β 0.21 [95%CI 0.05 to 0.36], *p* = 0.008) increased through time (Figure 1D). Serious adverse events such as myocardial infarction (β −2.30 [95%CI −4.05 to −0.56], *p* = 0.010) or stroke (β −1.51 [95%CI −3.34 to 0.31], *p* = 0.104) occurred less often in the later years of the investigated period.

### 3.3. Patients’ Characteristics of Nonsurvivors Compared with Survivors in Patients with ESRD and Psoriasis

Nonsurvivors were older than survivors of the in-hospital stay (72 [IQR, 62–77] vs. 66 [56–75] years, *p* < 0.001) and had more frequent comorbidities as coronary artery disease (46.6% vs. 30.2%, *p* = 0.013), COPD (32.8% vs. 15.2%, *p* < 0.001), or arterial fibrillation/flutter (46.6% vs. 21.1%, *p* < 0.001). We did not observe differences for other comorbidities. In nonsurvivors, left- ((48.3% vs. 25.7%), *p* < 0.001) and right-heart failure ((27.6% vs. 9.5%), *p* < 0.001) were more often present along with bleeding events such as gastrointestinal bleeding (5.2% vs. 0.3%, *p* < 0.001) with consecutively more frequent use of transfusions of erythrocytes concentrates (55.2% vs. 17.6%, *p* < 0.001)**.** In nonsurvivors, adverse events as myocardial infarction (6.9% vs. 0.8%), *p* < 0.001) and stroke (6.9% vs. 0.7%, *p* < 0.001) were more often diagnosed during hospitalization compared to survivors. Patients with ESRD and psoriasis with concomitant psoriatic arthritis did not show any difference in case-fatality rate, as opposed to patients without psoriatic arthritis (6.5% vs. 5.5%, ARR [95%CI] −0·6% [−3·1%; +7·4%]). Differences in baseline characteristics and comorbidities in those patients are presented in the Appendix A.

### 3.4. Impact of Psoriasis on Adverse Events on Patients with ESRD

Serious adverse events such as myocardial infarction (1.1% vs. 1.2%, *p* = 0.889) and stroke (1.0% vs. 0.6%, *p* = 0.113) were equally present in patients with ESRD and psoriasis compared to ESRD patients without psoriasis. Only pulmonary embolism tended to be more often diagnosed in patients with ESRD and psoriasis (0.6% vs. 0.2%, *p* = 0.051). The in-hospital mortality rate did not differ considerably between patients with ESRD and psoriasis compared to patients without psoriasis (5.5% vs. 5.9%, *p* = 0.554) (Table 1). The multivariate regression models demonstrated that the additional diagnosis of psoriasis in patients with ESRD was not associated with higher or lower risk of in-hospital mortality (OR 1.02 (95%CI 0.78–1.33), *p* = 0.915). Furthermore, no significant associations were detected for other events such as myocardial infarction (OR 0.84 (95%CI 0.48–1.49), *p* = 0.563), pulmonary embolism (OR 2.16 (95%CI 0.96–4.83), *p* = 0.062), or stroke (OR 1.64 (95%CI 0.90–2.97), *p* = 0.105). However, the additive diagnosis of psoriasis in ESRD patients was independently associated with an increased necessity regarding transfusions of erythrocyte concentrates (OR 1.18 (95%CI 1.02–1.38), *p* = 0.030) in patients with ESRD (Table 2).

## 4. Discussion

The main findings of our study can be summarized as follows:(I)Annual numbers of ESRD patients increased, whereas numbers of ERDS with psoriasis remained widely constant from 2010 to 2020.(II)While overall in-hospital mortality declined through time, patient characteristics shifted toward older age and more severe comorbidity profiles(III)Hospitalized ESRD patients with psoriasis were in median five years younger than in those without psoriasis.(IV)Despite younger age, we found a higher prevalence of severe, life-shortening comorbidities such as cancer and coronary heart disease in hospitalized patients with psoriasis than in patients without psoriasis.(V)The additive diagnosis psoriasis was not associated with the in-hospital case-fatality rate of patients with ESRD.

In our cohort of hospitalized ESRD patients, annual patient numbers generally increased during the 10-year interval from 2010 to 2020. In contrast, numbers of hospitalized ERDS patients with psoriasis remained widely constant annually in this period of time. Both results must be discussed in the context of the current literature: First, the increasing prevalence of ESRD in our cohort fits with international trends. ESRD is a growing disease with increasing patient numbers not only in industrialized but also/especially in developing countries [20]. However, regarding the observed annual downtrend of the admitted numbers of ESRD patients with concomitant psoriasis, the German data differ from previously published epidemiological data, demonstrating an increase in prevalence of psoriasis in the general population during the past years [21]. One explanation for this significant difference to our findings could be the way in which patient data were coded and collected. The coding of a secondary diagnosis in in-patients often does not take place in the case of concomitant diseases that are not obvious. Thus, especially patients with a mild manifestation of the disease who do not consider their disease as a further problem could be missing in our database. Additionally, it is conceivable that the improved therapeutic possibilities in recent years resulted in many milder courses [22,23] despite an increased number of psoriasis diagnoses. Furthermore, it is speculated that the increase in awareness and improved diagnostic options could have largely contributed to the increase in numbers of psoriasis diagnoses [24].

While solid evidence for the relation of severe psoriasis and cardiovascular comorbidity is given [25,26], the association between chronic kidney disease and psoriasis is contentious [3]. In our cohort, hospitalizations for CKD in patients with psoriasis occurred in median 5 years earlier in life than in CKD patients without psoriasis. This finding is highly alarming in itself—and it is consistent with epidemiological data showing a reduction in life expectancy by about the same amount of time in psoriasis patients [6]. Despite the younger age, the prevalence of coronary artery disease, heart failure, diabetes, and hypertensive nephropathy were significantly higher in our cohort of hospitalized ESRD patients with psoriasis. In line with the literature [3], our data indicate that psoriasis patients are at risk for developing chronic kidney disease together with all crucial cardiovascular risk factors and subsequent cardiovascular disease at earlier stages in their lifetime. We have previously shown that CKD stages 2, 3, and 5 were more common in a cohort of AKI patients with psoriasis, despite a significantly younger age of the patients—on average about 5 years as well [27]. With the current data, we are able to confirm these findings: psoriasis patients hospitalized with acute- or end-stage renal failure are significantly younger than patients in the same cohort without psoriasis.

Despite a higher prevalence of severe comorbidities in our cohort of ESRD patients with concomitant psoriasis, multivariate regression models showed that the additional diagnosis of psoriasis in patients with ESRD was not associated with changes for in-hospital mortality and severe cardiovascular events such as myocardial infarction, pulmonary embolism, or stroke. This finding is not in line with previously published work [28,29]. We provide the following suggestions that might explain this result: The significantly younger age of ESRD patients with psoriasis may be one of the main causes for the lower number of cardiovascular events and case-fatality rate (in median 5 years younger). Although the multivariate logistic regression model adjusted for age, sex, and important comorbidities showed no association of psoriasis and incidence rates of stroke, myocardial infarction, or in-hospital mortality, it has to be kept in mind that the model cannot compensate all age-related effects in comorbidities [25]. Additionally, the occurrence of cardiovascular events is not limited to the time in the hospital after diagnosis of ESRD, but can occur at any time afterward, which is not included in our study. Thus, patients can be of higher risk but it is not reflected by the registered time in the hospital.

Furthermore, we want to discuss other limitations of our study. As mentioned before, we cannot exclude that the reported results might be influenced by undercoding/underreporting of comorbidities (including psoriasis) in ESRD patients, as the data are based on ICD and OPS discharge codes of hospitalized patients. Especially patients who suffer from very mild forms psoriasis might not be coded as well as all patients who died immediately after admission when no full medical history was taken. This could result in lower survival rates in ESRD patients with psoriasis than reported. The prevalence of psoriasis in ESRD patients in our cohort was 0.3% and lower than the expected prevalence for psoriasis in industrialized nations (2–3%) [2], but was higher than in other studies [30]. Comedications are not coded in the database sufficiently and therefore the association between medications and in-hospital outcomes cannot be analyzed in the present study. Patients can be admitted to hospitals more than once during a year, which cannot be distinguished. Therefore, we report about the case-fatality. Finally, no classification of the extent of psoriatic skin disease is given and we were only able to study the association between variables detected during hospitalization, but cannot discuss the temporal or causal relationship.

Taken together, our study results should be considered hypothesis-generating, and large prospective observational studies are needed to further explore the impact of psoriasis on different stages of acute and chronic kidney disease. Nevertheless, we were able to analyze effects in a large study with a high number of ESRD patients who were coded during the observational period and therefore able to provide representative groups of ESRD patients with and without concomitant psoriasis. Additionally, we focused on baseline characteristics such as age and hard endpoints such as in-hospital mortality and in-hospital adverse events, which are rarely miscoded or not coded.

## 5. Conclusions

Our study data provide further insights into the triad of psoriasis, cardiovascular disease, and renal disease, and raise the awareness of this interaction of different comorbidities in psoriasis patients. Autoinflammatory skin and systemic psoriasis disease are accompanied by life-threatening comorbidities that take place outside the skin layers, putting heart and kidneys in danger and shortening the patient’s life. Awareness for these strong connections and interactions still needs be raised to foster a close cooperation between dermatology and other specialist disciplines such as nephrology and cardiology. This would improve and optimize management and treatment for these patients, focusing to normalize the lifespan of psoriasis patients.

## Figures and Tables

**Figure 1 jcm-11-04328-f001:**
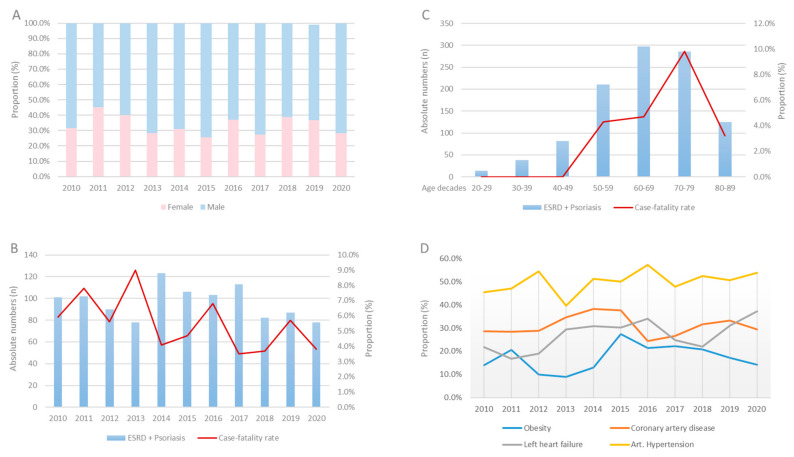
Proportion rate of gender distribution (**A**), absolute numbers, and case-fatality rate stratified across years (**B**) and decades (**C**) and proportion of comorbidities (**D**) in patients with ESRD and psoriasis between 2010 and 2020 in Germany.

**Table 1 jcm-11-04328-t001:** Baseline characteristics, treatment, and outcome of patients with end-stage renal disease (*n* = 360,980) stratified by concomitant psoriasis (cumulative data for the years 2005–2020). Bold means *p* < 0.05.

Parameters	ESRD with Psoriasis (*n* = 1063; 0.3%)	ESRD without Psoriasis (*n* = 359,917; 99.7%)	*p*-Value
**Age (years)**	**66 [56–75]**	**71 [59–79]**	**<0.001**
**Sex (female)**	**356 (33.5%)**	**144,561 (40.2%)**	**<0.001**
**Obesity**	**186 (17.5%)**	**29,610 (8.2%)**	**<0.001**
**In-hospital stay**	**8 (3–16)**	**5 (2–13)**	**<0.001**
**Comorbidities**			
**Coronary artery disease**	**331 (31.1%)**	**100,843 (28.0%)**	**0.026**
**Malignancy**	**52 (4.9%)**	**11,778 (3.3%)**	**<0.001**
**Left heart failure**	**286 (26.9%)**	**81,221 (22.6%)**	**0.001**
Right heart failure	111 (10.4%)	32,042 (8.9%)	0.082
**COPD**	**172 (16.2%)**	**48,906 (13.6%)**	**0.014**
**Diabetes mellitus**	**454 (42.7%)**	**138,502 (38.5%)**	**0.005**
Arterial hypertension	532 (50.0%)	172,778 (48.0%)	0.186
**Hypertensive Nephropathy**	**101 (9.5%)**	**26,325 (7.3%)**	**0.007**
Atrial fibrillation/flutter	239 (22.5%)	83,677 (23.2%)	0.584
Deep vein thrombosis or thrombophlebitis	9 (0.8%)	2906 (0.8%)	0.863
**Dialysis modalities**			
Dialysis general	467 (43.9%)	153,121 (42.5%)	0.368
Haemofiltration	26 (2.4%)	6662 (1.9%)	0.166
Haemodialysis	459 (43.2%)	150,567 (41.8%)	0.383
**Haemodiafiltration**	**108 (10.2%)**	**27,720 (7.7%)**	**0.003**
**Adverse events during hospitalization**			
Gastrointestinal bleeding	6 (0.2%)	3746 (1.0%)	0.169
Intracranial bleeding	0 (0%)	361 (0.1%)	0.632
Myocardial infarction	12 (1.1%)	4402 (1.2%)	0.889
Ischemic Stroke	11 (1.0%)	2248 (0.6%)	0.113
Pulmonary embolism	6 (0.6%)	888 (0.2%)	0.051
Cardiopulmonary reanimation	14 (1.3%)	5432 (1.5%)	0.693
**Transfusion of erythrocytes**	**209 (19.7%)**	**62,038 (17.2%)**	**0.037**
MACCE	73 (6.9%)	26,096 (7.3%)	0.673
In-hospital mortality	58 (5.5%)	21,362 (5.9%)	0.554

**Table 2 jcm-11-04328-t002:** Impact of psoriasis on different serious adverse events through hospitalization in patients with ESRD. Bold means *p* < 0.05.

	Univariate Regression Model		Multivariate Regression Model *	
	OR (95%CI)	*p*-Value	OR (95%CI)	*p*-Value
In-hospital mortality	0.92 (0.70–1.19)	0.509	1.015 (0.78–1.33)	0.915
Acute myocardial infarction	0.92 (0.52–1.63)	0.780	0.84 (0.48–1.49)	0.563
Ischemic Stroke	1.66 (0.92–3.02)	0.094	1.64 (0.90–2.97)	0.105
Deep venous thrombosis or thrombophlebitis	1.05 (0.54–2.02)	0.886	1.02 (0.53–1.97)	0.955
**Pulmonary embolism**	**2.30 (1.03–5.13)**	**0.043**	2.16 (0.96–4.83)	0.062
Gastrointestinal bleeding	0.54 (0.23–1.21)	0.132	0.54 (0.24–1.21)	0.137
**Transfusion of blood constituents**	**1.18 (1.01–1.37)**	**0.037**	**1.18 (1.02–1.38)**	**0.030**

* Adjusted for age, sex, cancer, coronary artery disease, chronic obstructive pulmonary disease, essential arterial hypertension, diabetes mellitus, and malignancy.

## Data Availability

The data that support the findings of this study are not publicly available due to data protection rules. Data are available as aggregated and only provided by the Federal Statistics Germany.

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
