# Peer review of "Case-Fatality and Temporal Trends in Patients with Psoriasis and End-Stage Renal Disease"

_jcm, 2022, doi:10.3390/jcm11154328_

Round 1

Reviewer 1 Report

Summary:

        The authors performed a study analyzing case-fatality and temporal trends in patients with psoriasis and end-stage renal disease based on the German nationwide inpatient sample. The article is well written and interesting, but there are still some points need to be clarified.

Major concern:

1.     Page 2 line 87. The authors did not include treatment modalities for patients with psoriasis in the analysis. However, some medications for psoriasis may be a factor associated with the major adverse cardiac events in these patients (Wu JJ, Joshi AA, Reddy SP, Batech M, Egeberg A, Ahlehoff O, Mehta NN. Anti-inflammatory therapy with tumour necrosis factor inhibitors is associated with reduced risk of major adverse cardiovascular events in psoriasis. J Eur Acad Dermatol Venereol. 2018 Aug;32(8):1320-1326).

2.     Page 2 line 91. The authors used ICD-10 code L40 to identify psoriatic patients. However, it is uncertain that whether there is any difference of fatality or comorbidities between psoriatic patients with and without psoriatic arthritis.

3.     Page 3 line 129. Were there any duplicate cases among the hospitalized patients included in your study? For example, some patients may admit to hospital several times for treatment of ESRD in the study period.

Minor concern:

1.     Page 3 table 1. The sum of cases of ESRD with and without psoriasis is not equal to 360,980. Please check it.

2.     Page 4 line 148. Although the case-fatality rate increased with age form the 3rd decade to the 8th decade, it decreased in patients over the age of 80.

3.     Page 6 table 2. Please explain the reason why patients with psoriasis are more likely to receive Transfusion of blood constituents than those without psoriasis in hospitalization.

Author Response

Response to the Reviewers´comments:

Response to the Reviewer #1:

REVIEWER 1:

Summary:

The authors performed a study analyzing case-fatality and temporal trends in patients with psoriasis and end-stage renal disease based on the German nationwide inpatient sample. The article is well written and interesting, but there are still some points need to be clarified.

RESPONSE: We would like to thank the Reviewer for his/her constructive criticism of our work. Below please find our point-by-point response. The changes made to address each one of your comments and suggestions are highlighted in yellow in the revised manuscript.

Major concern:

  1. Page 2 line 87. The authors did not include treatment modalities for patients with psoriasis in the analysis. However, some medications for psoriasis may be a factor associated with the major adverse cardiac events in these patients (Wu JJ, Joshi AA, Reddy SP, Batech M, Egeberg A, Ahlehoff O, Mehta NN. Anti-inflammatory therapy with tumour necrosis factor inhibitors is associated with reduced risk of major adverse cardiovascular events in psoriasis. J Eur Acad Dermatol Venereol. 2018 Aug;32(8):1320-1326).

RESPONSE: We totally agree with this comment that the inclusion of specific medication would improve the manuscript. Specific co-medications are coded in the German nationwide sample very rarely. In the context of psoriasis, the German nationwide did not include enough cases of co-medications to include such an analysis. We added a sentence in the limitation paragraph to clarify this point on page 13, line 7-9.

Co-medications are not coded in the database sufficiently and therefore possible drug-related side effects could be part of our study.”

  1. Page 2 line 91. The authors used ICD-10 code L40 to identify psoriatic patients. However, it is uncertain that whether there is any difference of fatality or comorbidities between psoriatic patients with and without psoriatic arthritis.

RESPONSE: Thank you very much for this important suggestion. As recommended, we performed a subanalysis regarding ESRD patients with psoriatic arthritis compared to patients without psoriatic arthritis. First, we did not find any difference on mortality between both groups. This is reported now in the manuscript on page 8, line 7-11.

In patients with ESRD and psoriasis with concomitant psoriatic arthritis did not show any difference in case-fatality rate opposed to patients without psoriatic arthritis (6.5% vs. 5.5%, ARR [95% CI] -0·6% [-3·1%; +7·4%]).

Second, we added a table comparing both patients groups regarding baseline characteristic and comorbidities in a new Table S1 in the supplementary material and refer to this table in the manuscript on page 8, line 10-11.

Differences in baseline characteristics and comorbidities in those patients are presented in the supplementary material.

  1. Page 3 line 129. Were there any duplicate cases among the hospitalized patients included in your study? For example, some patients may admit to hospital several times for treatment of ESRD in the study period.

RESPONSE: Thank you for this important remark. Indeed, the German nationwide sample cannot distinguish between re-admissions. Thus, some patients may admit several times. For this reason we report about the case-fatality rate. We added this fact to the limitation section in the manuscript on page 11, line 9-10:

Patients can be admitted to hospitals more than once during a year, which cannot be distinguished. Therefore, we report about the case-fatality.”

 Minor concern:

  1. Page 3 table 1. The sum of cases of ESRD with and without psoriasis is not equal to 360,980. Please check it.

RESPONSE: We corrected this mistake.

  1. Page 4 line 148. Although the case-fatality rate increased with age form the 3rddecade to the 8th decade, it decreased in patients over the age of 80.

RESPONSE: Thank you for this suggestion. We added such a sentence to the manuscript in the results section on page 7, line 20-22.

However, although the case-fatality rate increased with age form the 3rd decade to the 8th decade, it decreased in patients over the age of 80 (Figure 1B and 1C).

  1. Page 6 table 2. Please explain the reason why patients with psoriasis are more likely to receive Transfusion of blood constituents than those without psoriasis in hospitalization.

RESPONSE: Since the reasons for the transfusions are not included in the German nationwide sample, we can only discuss the observed higher transfusion frequency in the context of the published literature. Our data show that ESRD patients with psoriasis had more often coronary heart disease or malignancies than ESRD patients without psoriasis. These co-morbidities might be major drivers for increased rates of erythrocyte transfusions in ESRD patients with psoriasis. There is still a controversy in the field of cardiologists regarding transfusion goals in patients with coronary artery disease. As it is clinical practice, to set hemoglobin levels above 7–8 g/dL as transfusion goal in most patients, there is data for patients with coronary artery disease to benefit from higher hemoglobin levels (PMID: 23708168), which could be one reason for a higher amount of erythrocyte transfusions in these patients. In quite the same way, anemia is one of the most frequent complication in patients with malignancies receiving chemotherapy – so these patients do need transfusions more frequently. 

Reviewer 2 Report

The article titled ‘Case-fatality and temporal trends in patients with psoriasis and end-stage renal disease’ may be an useful contribution to the journal; however, few changes should be taken into consideration:

Authors conclude that ‘Multivariate regression models demonstrated that psoriasis did not impact on in-hospital case-fatality in patients with ESRD (OR 1.02 (95%CI 0.78-1.33), P=0.915)’  This is not sustained, as such thing cannot be inferred from the current analysis; at most, it can be sustained that the ‘multivariate regression models did not demonstrate that psoriasis impacted on in-hospital case-fatality in patients with ESRD’, as the present research as it was tailored could not confirm the null hypothesis, it only failed to reject it, an aspect we consider to be a small difference of paramount importance in the context of the manuscript.

Figure 1 legend- at chart C must be specified “age decades”, to be more specific.

Obesity and Hypertension should be defined in Methods, for conformity reasons; also, shuld be mentioned if hypertension values changed in definition as guidelines for diagnose of arterial HT may fluctuate in values worldwide with time.

Grammar and punctuation must also be carefully checked within the entire article  (e.g. more parantheses  such as in line 32, and also more).

Author Response

Response to the Reviewers´comments:

Response to the Reviewer #2:

REVIEWER 2:

The article titled ‘Case-fatality and temporal trends in patients with psoriasis and end-stage renal disease’ may be an useful contribution to the journal; however, few changes should be taken into consideration:

RESPONSE: We would like to thank the Reviewer for his/her constructive criticism of our work. Below please find our point-by-point response. The changes made to address each one of your comments and suggestions are highlighted in yellow in the revised manuscript.

Authors conclude that ‘Multivariate regression models demonstrated that psoriasis did not impact on in-hospital case-fatality in patients with ESRD (OR 1.02 (95%CI 0.78-1.33), P=0.915)’  This is not sustained, as such thing cannot be inferred from the current analysis; at most, it can be sustained that the ‘multivariate regression models did not demonstrate that psoriasis impacted on in-hospital case-fatality in patients with ESRD’, as the present research as it was tailored could not confirm the null hypothesis, it only failed to reject it, an aspect we consider to be a small difference of paramount importance in the context of the manuscript.

RESPONSE: We totally agree with the Reviewer on the aspect that the conclusion regarding the impact of psoriasis should be more carefully worded. In this context we revised this wording throughout the manuscript in the abstract on page 3 (line 18), in the results section on page 8 (line 20-23) and in the discussion section on 9 (line 12).

Multivariate regression models demonstrated that psoriasis were not associated with in-hospital case-fatality in patients with ESRD (OR 1.02 (95%CI 0.78-1.33), P=0.915).

The multivariate regression models demonstrated that the additional diagnosis of psoriasis in patients with ESRD was not associated with higher or lower risk of in-hospital mortality (OR 1.02 (95%CI 0.78-1.33), P=0.915).

V) The additive diagnosis psoriasis were not associated with the in-hospital case-fatality rate of patients with ESRD.

Figure 1 legend- at chart C must be specified “age decades”, to be more specific.

RESPONSE: Thank you very much for your comment. As recommended, we added “age decades” to the Figure 1 at chart c.

Obesity and Hypertension should be defined in Methods, for conformity reasons; also, shuld be mentioned if hypertension values changed in definition as guidelines for diagnose of arterial HT may fluctuate in values worldwide with time.

RESPONSE: Thank you very much for this interesting suggestion.

First, we added to method section the definition for obesity on page 5, line  25-26, which is available in the German nationwide sample:

“ […] essential arterial hypertension [ICD-code I10] and obesity [defined as body mass index [BMI] ≥30 kg/m² according to the World Health Organization; ICD code E66] […]

Second, the German nationwide sample does not include a clear definition for hypertension. Furthermore, exact hypertension values are not part of the German database. Thus, we unfortunately cannot report about changes regarding guidelines.

Grammar and punctuation must also be carefully checked within the entire article (e.g. more parantheses  such as in line 32, and also more).

RESPONSE: We thoroughly revised our manuscript to correct grammar and punctuation typos.

Round 2

Reviewer 1 Report

Summary:

In the revised version, the authors adequately respond to the point raised by me. However, I suggest the authors do some minor revisions.

1. Page 3 Table 1. Please correct the Table legends (n=360,989) to (n=360,980).

2. Page 7 line 267. Consider revising the sentence as “Co-medications are not coded in the database sufficiently and therefore the association between medications and in-hospital outcomes cannot be analyzed in the present study.” Since some medications may reduce the risk of major adverse cardiovascular events in such patients.

Author Response

Revision 2: jcm-1762741

Response to the Reviewer #1:

REVIEWER 1:

In the revised version, the authors adequately respond to the point raised by me. However, I suggest the authors do some minor revisions.

RESPONSE: We would like to thank the Reviewer for his/her positive feedback of our work. Below please find our point-by-point response. The changes made to address each one of your comments and suggestions are highlighted in yellow in the revised manuscript.

Page 3 Table 1. Please correct the Table legends (n=360,989) to (n=360,980).

RESPONSE: We corrected this mistake.

Page 7 line 267. Consider revising the sentence as “Co-medications are not coded in the database sufficiently and therefore the association between medications and in-hospital outcomes cannot be analyzed in the present study.” Since some medications may reduce the risk of major adverse cardiovascular events in such patients.

RESPONSE: As recommended by the reviewer we corrected the sentence on page 11, line 9-11 as the following:

Co-medications are not coded in the database sufficiently and therefore the association between medications and in-hospital outcomes cannot be analyzed in the present study.”

Reviewer 2 Report

Manuscript had been improved.  

Author Response

Revision 2: jcm-1762741

Response to the Reviewer #2:

REVIEWER 2:

Manuscript had been improved.

RESPONSE: We would like to thank the Reviewer for his/her positive feedback of our work.